# Effect of Dietary Folic Acid Supplementation during Pregnancy on Blood Characteristics and Milk Composition of Ewes

**DOI:** 10.3390/ani10030433

**Published:** 2020-03-04

**Authors:** Bo Wang, Zhen Li, Heqiong Li, Hailing Luo, Hugh T. Blair, Luyang Jian, Zhicheng Diao

**Affiliations:** 1State Key Laboratory of Animal Nutrition, College of Animal Science and Technology, China Agricultural University, Beijing 100193, China; wangboforehead@163.com (B.W.); lizhen6394@126.com (Z.L.); joan374261795@163.com (H.L.); jianluyang@cau.edu.cn (L.J.); diaozhicheng@cau.edu.cn (Z.D.); 2Sheep Research Group, School of Agriculture and Environment, Massey University, Private Bag 11222, Palmerston North 4442, New Zealand; h.blair@massey.ac.nz; 3National Research Centre for Growth and Development, Massey University, Private Bag 11222, Palmerston North 4442, New Zealand

**Keywords:** folic acid, gestation period, litter size, serum, milk composition, ewe

## Abstract

**Simple Summary:**

Folic acid (FA) plays a critical role in regulating fetal development. Cause of the association between maternal metabolism and fetal development has created interest to study FA’s effects on maternal blood metabolism and milk composition. FA was supplemented in the diet of ewes with different litter size during pregnancy and blood parameters, and milk composition was monitored. The results indicated that FA supplementation improved folate metabolism balance during gestation of prolific ewes, and contributed to the ewe’s blood metabolism and health. In addition, immune-related compounds in milk were improved with dietary FA supplementation, thus, the newborn lamb should benefit from the improvement in milk quality.

**Abstract:**

The objective of the present study was to investigate the dynamic change of serum parameters and milk composition by dietary FA supplementation with ewes with different litter size from mating to lambing. The ewes were divided into six treatments (TW-CON, TW-F16, TW-F32, TR-CON, TR-F16, TR-F32) according to dietary FA levels (control, CON; 16 or 32 mg·kg^−1^ rumen-protect-FA supplementation, F16 and F32) and litter size (twin born, TW; and triplet born, TR). In serum, the concentration of folate increased linearly with dietary FA supplementation (*P* < 0.05), regardless of the litter size, they showed a quadratic response to gestation progression (*P* < 0.05). With dietary FA addition, IGFI-I levels significant increased from late gestation to after lambing (*P* < 0.05), and linearly increased immunoglobulin during the perinatal period (*P* < 0.05). In colostrum and milk at d 15, the content of folate, lactoferrin, and IgG were affected positively by FA supplementation (*P* < 0.05). IgG was higher in the TW group than TR in colostrum (*P* < 0.05), and lactoferrin in TW was lower than TR in milk of d 15 (*P* < 0.05). FA supplementation increased protein content in colostrum (*P* < 0.05), while it had no effect on the fat, lactose, and BUN of colostrum and milk of d 15 (*P* > 0.05). These results suggest that FA supplementation during gestation could regulate maternal blood metabolism and contribute to milk immune composition.

## 1. Introduction

As an essential micronutrient, folic acid (FA) plays a crucial role in various metabolic processes of the cell, and it is necessary to maintain and improve fetal body development during pregnancy in humans and animals. The positive effects of FA supplementation on birth weight is well documented [1]. Its benefit for animal growth may be because adequate FA intake is required for cell division and homeostasis due to its essential role as a coenzyme in nucleic acid synthesis and repair [2,3]. FA also regulates mitochondrial biogenesis and function through changing the mitochondrial DNA content and gene expression levels to compensate tissue impairment [4]. FA contributes to increased antioxidant capacity in rats [5] and immunity in freshwater prawns [6]. Collectively, these functions suggest that FA is necessary in regulating the fetus or young animals’ growth, development, and metabolism processes. We speculate that the maternal metabolism could also be modulated by dietary FA ingestion and this may affect the offspring’s metabolism, especially in the early stages of growth.

Due to genetic improvement and intensification in past decades, the reproduction performance in prolific breeds of sheep and the lactation performance of cows has increased substantially. This increased performance may result in an increased demand for FA to maintain higher production. In ruminants, rumen microorganisms synthesize folate. Several studies have demonstrated that dietary FA (unprotected and protected) supplementation increased milk production, protein concentration, and milk yield of dairy cows [7,8,9,10]. The level of folates supplied by diet and ruminal microbes is not sufficient to maintain the serum folate concentrations at a constant level throughout gestation, and the decrease of serum concentrations is greater in prolific breeds from mating to 60 days of gestation of ewes [11]. Collectively, these results suggest that FA requirements are increased during gestation and lactation in high performing females. 

Haematochemical parameters are good indicators of the physiological and health conditions in animals [12,13]; and the quality of milk in ewe is critical for neonatal lambs, especially for the acquisition of passive immunity [14]. Most previous studies focused on fetal growth or dam milk production, few studies have reported the effects of FA supplementation on maternal metabolism in ewes. 

The aim of the present study was undertaken to determine the effect of protected dietary FA supplementation throughout the gestation period on the time-course of serum parameters from mating to lambing and milk components in prolific ewes with twin or triplet fetuses.

## 2. Material and Methods

### 2.1. Animals and Experiment Design

Procedures throughout this experiment were in accordance with the Animal Ethics Committee of Beijing. The protocol used throughout the study was approved by the Institutional Animal Care and Use Committee of the China Agricultural University (Permit number: DK996).

One hundred and twenty *Hu* sheep with similar body weights (44.00 ± 0.39 kg), and multiparous ewes (all animals had given birth twice before, 24 ± 4.2 months of age) and showing signs of estrus were selected, and randomly divided into three treatments after mating. Ewes were housed in individual pens (size: 1.5 × 3 m^2^) and given one of the three rations throughout the gestation period: control (no FA supplementation, CON); supplemented with 16 mg (F16) or 32 mg (F32) rumen-protected FA per kilogram dry matter (DM). The specific parameters of supplemented FA in this trial were the following: purity, 99.8%; rumen passing rate and small intestinal absorption rate (measurement by rumen fistula and small intestine fistula), 92.60% and 85.59%, respectively. Twenty-eight days after mating, type-B ultrasonography was used to check whether the ewes were pregnant or not. Non-pregnant ewes were removed from the groups; the litter size was not identified until lambing. After lambing, according to the litter size at birth (Twin-born, TW; Triplet-born, TR) and the ingested FA levels in diet, the ewes were divided into 6 groups (TW-CON n = 16, TW-F16 n = 13, TW-F32 n = 10, TR-CON n = 5, TR-F16 n = 13 and TR-F32 n = 9) as a 2 × 3 factorial design. 

The dietary component and nutrient levels were provided according to the recommendations for small ruminants of the National Research Council 2007 [15]. The percentage of DM basis of peanut vines, whole corn silage, and concentrate in total mixed ration (TMR) from early (from mating to 90 days) to late (from day 91 to lambing) gestation was 50.00 : 45.00 : 5.00 and 27.00 : 28.00 : 45.00, respectively. Concentrate composition and nutrient levels of TMR in the control group are shown in Table 1. Folic acid was supplemented to the total mixed ration for each of the individual ewes, the amount of TMR offered changed gradually throughout the gestation period to meet the NRC recommendation, and fed twice daily and equally at 8:00 am and 6:00 pm. The ewes had free access to drinking water throughout the experiment.

### 2.2. Sampling Procedure and Measurement

#### 2.2.1. Dietary Chemical Analyses

TMR samples were collected every two weeks during early (from mating to 90 days of gestation) and late (from 91days to parturition) gestation period and dried in an oven at 65 °C for 48 h to obtain air-dry basis matter. Samples were then ground and passed through a sieve with opening size of 1 mm mesh for chemical analyses. Dry matter was measured by drying the samples in a 105 °C oven for 2 h, content of ash was assessed by burning the samples in a 550 °C muffle furnace until reaching a constant weight. Crude protein was determined by nitrogen content (measured by macro-Kjeldahl method) multiplied by 6.25 [16]. A reflux system (ANKOM XT15, Ankom Technology, Macedon, NY, USA) with petroleum ether was performed for 1 hour at 90 °C to detect the ether extract content in diets. Neutral and acid detergent fiber in the diets were measured following the methods as described by Van Soest et al. [17]. An atomic absorption spectrometer (Czerny-Turner AAS8000, Skyray Instruments, Dallas, TX, USA) was used to determine the content of calcium and the molybdenum blue colorimetric method was performed to measure the phosphorus content [16].

#### 2.2.2. Serum Parameters

Blood samples were collected from the jugular vein through a 10 mL vacuum tube in the morning before feeding at day 1, 30, 90, 140 of pregnancy and the first day after parturition. The serum samples were separated by centrifugation of blood and stored at −20 °C for determination of serum variables. The concentration of folate, homocysteine (Hcy), insulin-like growth factor-I (IGF-I), and growth hormone (GH) were determined using a sheep enzyme-linked immunosorbent assay (ELISA) commercial kit (DuMa Biological Technology Development Co., Ltd., Shanghai, China), following the manufacturer’s instructions. The content of immunoglobulin (IgG, IgM, IgA) in the serum were detected by a KHB-1280 automatic biochemical analyzer (KeHua Bio-engineering Co., Ltd., Shanghai, China).

#### 2.2.3. Milk Composition

Colostrum samples were collected at 12 and 36 h after lambing, and preserved with 2-bromo-2-nitropropane-1, 3-diol. The two samples were sub-sampled and mixed together proportionately to each sample weight for milk composition determination. Milk samples were taken in the morning of days 14, 15, and 16 after parturition, and mixed proportionately as for the milk samples of day 15. The milk samples were preserved the same as colostrum. Concentrations of folate and lactoferrin in the colostrum and milk were measured following the instruction of a sheep enzyme-linked immunosorbent assay (ELISA) commercial kit (DuMa Biological Technology Development Co., Ltd., Shanghai, China). IgG content in the colostrum and normal milk were conducted as the manufacturer’s instruction of the ELISA commercial kit (DuMa Biological Technology Development Co., Ltd., Shanghai, China). The content of protein, fat, lactose, and BUN in the colostrum and normal milk were detected by milk analyzer (Milko-Scan FT +, Foss Electric, Hillerød, Denmark).

### 2.3. Statistical Analysis

Indices of blood and milk were analyzed using the generalized liner model (GLM) procedure of SPSS version 22.0 (SPSS, IBM, Inc., Chicago, USA) to assess the main effects of litter size (TW and TR) and the FA levels (CON, F16 and F32), and the interaction effects between the litter size and FA levels. Statistical significance of the main effect was demonstrated at a level of *P* < 0.05. The time-course effect (from mating to lambing) of folate, Hcy, IGF-I, and GH in the serum of each treatment was analyzed by generalized linear mixed model (GLMM) using SPSS. Polynomial analysis was performed to test the linear or quadratic response of variables to the dietary FA levels or the time-course effect.

## 3. Results

### 3.1. Serum Variables from Gestation to Parturition

#### 3.1.1. Folate and Hcy

No significant interaction effects (litter size × FA levels) on serum folate and Hcy concentrations were found in this experiment (*P* > 0.05; Table 2). The serum folate concentration was higher in the TW group than TR only at 90 days after mating (*P* < 0.05), while the concentration showed a significant linear increase with dietary FA supplementation at 1, 30, 90, and 140 days of gestation and after lambing (*P* < 0.05). For the time-course effect, serum folate concentrations in the six treatments exhibited a quadratic response throughout the gestation period, while decreasing from mating to the lowest point at 90 days of pregnancy, and then increasing until parturition (*P* < 0.05, Figure 1A). 

No significant difference was found between the TW and TR group on Hcy concentrations on the 30th, 90th, and 140th day of pregnancy and after lambing (*P* > 0.05). Concentrations of Hcy decreased linearly with FA addition in the diets (*P* < 0.05; Table 2). Concentrations of Hcy were non-significant throughout gestation in each of the 6 treatments (*P* > 0.05; Figure 1B).

#### 3.1.2. IGF-I and GH

Concentration of IGF-I and GH in the serum did not differ significantly between TW and TR groups throughout the gestation period (*P* > 0.05; Table 3). On the 90th of gestation and after lambing, IGF-I levels increased linearly with dietary FA supplementation (*P* < 0.05; Table 3); with FA supplementation in the TR groups (TR-F16 and TR-F32), the IGF-I also indicated a linear increase with the progression of pregnancy (*P* < 0.05), but did not change significantly in other treatments (*P* > 0.05; Table 4). The concentration of GH at different time among the treatments was not affected by litter size and dietary FA levels (*P* > 0.05; Table 3), while the concentration increased linearly as gestation progressed until lambing (*P* < 0.05; Table 4).

#### 3.1.3. Immunoglobulin

Compared with ewes ingested control diet, IgG content increased linearly with FA supplementation in the diet at 140 days of pregnancy and after lambing (*P* < 0.05); and postpartum IgG showed a higher level in the TR group than TW (*P* < 0.05; Table 5). Either in the TW or TR group, IgG concentration indicated a linear increase from mating to laming with dietary FA supplementation (*P* < 0.05), while no significance was found in ewes the fed control diet (*P* > 0.05; Table 6). Concentrations of IgM and IgA only exhibited a linear increase after lambing in the ewes whose diet was supplemented with FA (*P* < 0.05) (Table 5). From 30 days after pregnancy to lambing, the concentrations of IgM exhibited a linear increase in the TR groups with FA supplementation (*P* < 0.05), but the levels in others treatments stand in the same line throughout the experiment (*P* > 0.05). Concentrations of IgA were not significantly different in each of the treatments throughout gestation until lambing (*P* > 0.05; Table 6).

### 3.2. Milk Components

#### 3.2.1. Folate, IgG and Lactoferrin Concentrations

Folate concentrations in both colostrum and milk of the 15th day were not influenced by litter size (*P* > 0.05; Figure 2A), but it increased linearly in colostrum in response to dietary FA supplementation, and a similar tendency was found at milk from the 15th day (*P* < 0.05; Figure 2B). The colostrum IgG content showed a higher level in TW than that in TR group (*P* < 0.05; Figure 3A), whereas no difference was observed in milk after at 15 days of lambing (*P* > 0.05; Figure 3C). Dietary FA supplementation also linearly increased the IgG levels in the colostrum and the 15th days milk (*P* < 0.05) (Figure 3B,D). Lactoferrin content in colostrum was not affected by litter size (*P* > 0.05). However, it showed a higher level in TR group when compared with its content in TW at 15 days after lambing (*P* < 0.05; Figure 4A). The concentration of lactoferrin increased in both colostrum and milk on the 15th day in response to the FA supplemented in diets (*P* < 0.05; Figure 4B).

#### 3.2.2. Colostrum and Normal Milk Composition

In both colostrum and the milk at 15 days after parturition, the FA levels and litter size show no significant influence on the concentrations of fat, lactose and BUN (*P* > 0.05; Table 7). Whereas protein content increased linearly (*P* < 0.05) in colostrum with maternal diet FA supplementation, and it still tended to be higher from the 15th day’s milk from ewes with FA supplemented during gestation period than the milk form ewes fed the control diet (*P* = 0.098).

## 4. Discussion

Serum folate concentration is an accurate indicator of the status of body folate [18]. In the present study, serum concentration of folate increased linearly with dietary FA supplementation throughout the experiment, which is consistent with previous studies in dairy cows [10,19,20] and sheep [21]. The folate concentration was higher in twin born ewes than triplet born ewes at 90 days after mating, this may be due to the higher folate requirement as gestation progressed with larger litter size from mating to early gestation. The result is similar to a previous report that serum folate concentration decreased more between mating and 60 d of gestation in prolific sheep breeds than for non-prolific breeds [11]. Regardless of dietary FA supplementation, we found the pattern of serum folate concentration had the same trend from mating to lambing, decreasing from mating to the beginning of late gestation period, then increasing until laming. These changes may be related to the increased requirement of FA in early gestation for embryo development and fetus organ formation, and then decreased fetal requirement after organ formation in late gestation. Similarly, Girard et al. (1996) reported that folate concentration of ewes from different breeds with different litter size showed a quadratic response to time to pregnancy (from mating to 140 d of gestation) [11]. The lowest folate content at 90 d of gestation with dietary FA supplementation was still higher than that of the early gestation period of ewes fed the control diet. Thus, we considered the negative balance of maternal folate during the gestation period has been improved by maternal dietary FA addition. 

Hcy concentrations decreased with FA supplementation in the diet compared with the control groups from 30 days of pregnancy to lambing, the decreased Hcy levels in serum are consistent with studies in dairy cows [10] and piglets [22]. Dietary folate depletion could result in an increase of serum Hcy, which could promote oxidative stress [5]. Maternal dietary FA supplementation improved serum folate concentration and decreased Hcy concentrations from gestation to lambing. Overall concentrations of folate and Hcy were consistent among all the treatments regardless of litter size and supplementary FA levels.

IGF-I and GH are of major significance in the regulation of cell metabolism, muscle development, adipogenesis, glucose, and energy metabolism [23,24,25]. Therefore, IGF-I and GH are valuable indicators for evaluating the animal’s metabolic condition. In the current study, IGF-I concentrations in the serum increased with FA supplementation from late gestation to lambing regardless of litter size, which means FA had a positive effect on maintaining maternal metabolism. However, as gestation progressed, IGF-I levels only in the ewes fed additional FA and pregnant with triplet fetuses increased linearly. We speculated that this might be due to dietary FA supplementation maintaining maternal and fetal nutrients metabolism by regulating maternal IGF-I levels to satisfy the higher requirement of nutrients for ewes pregnant with three fetuses. In addition, maternal serum IGF-I also plays a key role in enhancing the placental function in sheep [26], while folate is critical in modulating placental nutrient transport and fetal growth [27]. Thus, it is likely that FA influences the fetus’s development by regulating maternal IGF-I levels to stimulate placental efficiency, which deserves further study. 

GH was not affected by litter size or dietary FA supplementation, and the linear increase of GH from early gestation to lambing in the ewes pregnant with twins with FA supplementation and the ewes pregnant with triplets. Maternal GH concentrations increased as gestation progressed presumably to modulate the growth of the fetus [28]. 

We also considered the effect of FA on immunity because of its critical role on health. Supplementary FA enhances the immune system in aquatic animals [6,29] and humans [30]. The increased immunoglobulin in maternal serum offer FA supplementation in the perinatal period could contribute to promoting the ewe’s immunity during perinatal period and postpartum recovery. From early gestation to lambing, the increase of IgG in both TW and TR ewes indicated the ewe’s immunity was positively affected by dietary FA levels during pregnancy. Results in human (long-term synthetic FA supplementation) [30] and broiler (in ovo injection of FA) [31] also showed FA supplementation was beneficial to immune function. IgM only increased in TR ewes treated with FA addition, the lack of response by TW ewes during gestation remains unclear and is deserving of further research.

There is a close relationship between milk composition and blood constituents [32,33]. In the present study, folate concentrations in colostrum increased with dietary FA supplementation during gestation. It is possible that the increasing blood folate from late gestation to lambing may contribute to the preparation of the mammary gland for lactation. Girard et al. (2005) reported that folate concentration in milk of cows showed a quadratic response to different dietary FA supplementation, and this may be because milk folate secretion is regulated by the mammary gland [34]. In the current study, milk folate concentration at day 15 of lactation was still higher in ewes with FA supplementation than CON ewes. This means maternal FA supplementation during pregnancy has a long-term influence on milk folate concentration.

Colostrum is a nutrient-rich source with complex biological components and contributes to the initial acquired immunity of neonates [35]. Colostrum IgG as the major immunoglobulin involved in stimulating the maturation of the newborn’s immune system [36]. Lactoferrin not only plays a role as an iron-binding protein with bacteriostatic properties, but also works as a host defense mechanism against microbial infection [37]. The IgG and lactoferrin content in both colostrum and milk increased with dietary FA supplementation, which means maternal FA supplementation during pregnancy may contribute to the immunity of newborn lambs. With FA supplemented in the diet, the increased IgG content in the colostrum and milk consist with the variance of serum IgG concentrations in the late gestation and lactation, and previous study demonstrated the positive relationship of IgG concentrations between the ewe’s serum and colostrum [38]. We also found IgG content in colostrum was higher in TW than TR. This may be explained by the litter size having a positive relationship with milk production [39] and milk production having a negative relationship with milk components [40]. Compared with colostrum, IgG and lactoferrin content were decreased in milk at day 15 of lactation. Ewes with triplets produced milk with higher concentration of lactoferrin. We suspect that mothers with a larger litter size could stimulate more lactoferrin secretion to protect the newborn lambs against bacteria and viruses from the environment to ensure better survival of their offspring.

Folic acid is an essential nutrient for protein synthesis, and contributes to the optimization of milk quality [20]. For multiparous cows, folic acid injection or dietary supplementation increases milk protein [34,41]. In our study, the protein content in colostrum also increased with FA supplemented in the diet during pregnancy, and showed a tendency to contain more protein in milk at day 15 of lactation. It is hypothesized that FA supplementation influences lactation performance by changing the regeneration of methionine from homocysteine, thereby regulating the milk protein [34]. Other studies suggest supplementary FA may modulate lactational performance by improving energy balance [10] or energy metabolism [41]. In addition, Sacadura et al. (2008) reported that vitamin B supplementation increased milk and milk component yields probably due to improved metabolic efficiency of intermediary metabolism [42]. 

Therefore, FA supplementation during gestation could contribute to maternal blood metabolic and improve milk quality, which suggested its benefit for the growth and development of offspring [43,44]. Moreover, more research is needed to investigate maternal nutrient digestion and transport with FA supplementation during gestation, and milk quality affected by FA supplementing in lactation diet, especially in high prolific breeds.

## 5. Conclusions

Our results indicate dietary folic acid supplementation increases serum folate concentration to improve folate metabolism balance during the gestation period of prolific ewes. Moreover, the various Hcy, IGF-I, GH, and immunoglobulin concentrations influenced by supplementary folic acid in the diet and gestation progress, especially in the ewes with triplet litters and in the period from late gestation to parturition, suggest folic acid may contribute to pregnant ewes metabolism and health. In addition, supplementary folic acid improved the quality of colostrum and milk by modifying their concentrations of folate, lactoferrin, IgG, and protein. 

## Figures and Tables

**Figure 1 animals-10-00433-f001:**
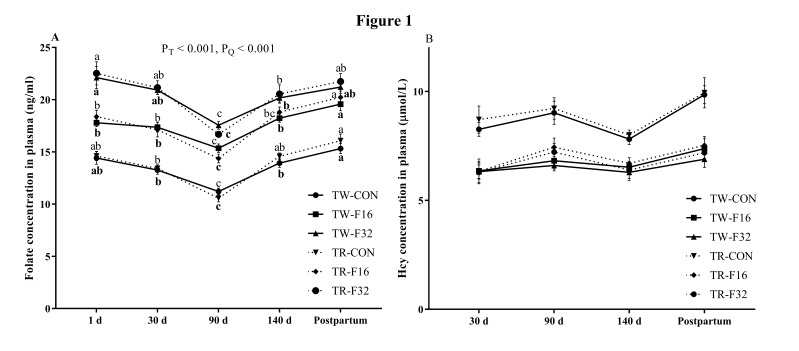
Folate (**A**) and Hcy (**B**) concentration in serum. Data are means ± SEM. Hcy, homocysteine; TW-CON, TW-F16 and TW-F32, twins born from ewes fed 0, 16, or 32 mg·(kg.DM)^−1^ FA in the basal diet, respectively; TR-CON, TR-F16 and TR-F32, triplets born from ewes fed 0, 16, or 32 mg·(kg.DM)^−1^ FA in the basal diet, respectively. *P*_T_, *P* value of time effect; *P*_Q_, *P* value of quadratic effect. Different lowercases indicated significance of corresponding TR groups (dotted line); different bold lowercases indicated significance of corresponding TW groups (full line). N = 5 for TR-CON, and n = 6 for other groups.

**Figure 2 animals-10-00433-f002:**
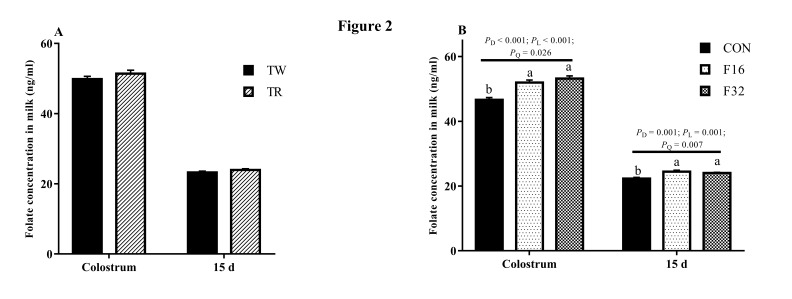
Effect of litter size (**A**) and FA levels (**B**) on the folate concentrations in milk. Data are means ± SEM. TW, twin born; TR, triplet born; CON, F16 and F32, ewes fed 0, 16, or 32 mg·(kg.DM)^−1^ FA in the basal diet, respectively. *P*_D_, *P* value of diet FA supplementation; *P*_L_, *P* value of liner effect; *P*_Q_, *P* value of quadratic effect.

**Figure 3 animals-10-00433-f003:**
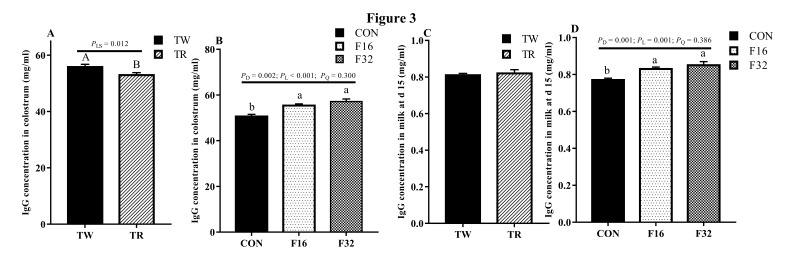
IgG concentrations affected by litter size (**A**,**C**) and dietary FA levels (**B**,**D**) in milk. Data are means ± SEM. TW, twin born; TR, triplet born; CON, F16 and F32, ewes fed 0, 16, or 32 mg·(kg.DM)^−1^ FA in the basal diet, respectively. *P*_LS_, *P* value of litter size; *P*_D_, *P* value of diet FA supplementation; *P*_L_, *P* value of liner effect; *P*_Q_, *P* value of quadratic effect.

**Figure 4 animals-10-00433-f004:**
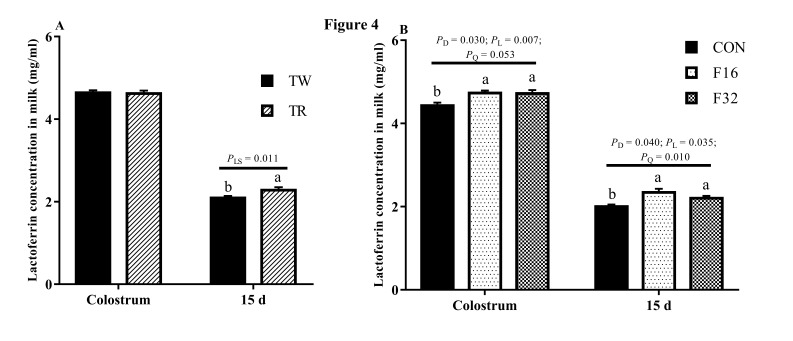
The concentration of lactoferrin influenced by litter size (**A**) and FA levels (**B**) in milk. Data are means ± SEM. TW, twin born; TR, triplet born; CON, F16 and F32, ewes fed 0, 16, or 32 mg·(kg.DM)^−1^ FA in the basal diet, respectively. *P*_LS_, *P* value of litter size; *P*_D_, *P* value of diet FA supplementation; *P*_L_, *P* value of liner effect; *P*_Q_, *P* value of quadratic effect.

**Table 1 animals-10-00433-t001:** Dietary concentrate components and nutrient levels of the basal diet (dry matter basis, %).

Ingredients	Formula of Concentrate			TMR Nutrient Levels	
	EG	LG	Nutrients	EG	LG
Corn	53.00	50.00	Dry matter	92.35	90.96
Soybean meal	9.70	22.50	Crude protein	9.60	10.00
Rapeseed meal	12.00	7.00	Ether extract	3.51	4.52
Wheat bran	15.70	11.50	Ash	11.19	8.85
Limestone	1.00	1.00	Neutral detergent fiber	49.19	34.15
CaHPO_4_	0.60	0.60	Acid detergent fiber	29.98	24.86
NaHCO_3_	1.30	1.30	Metabolizable energy ^2)^	1.93	2.15
NaCl	0.80	0.40	Calcium	0.54	0.56
Vitamin E	0.40	0.10	Phosphorus	0.37	0.26
Soybean oil	0.30	0.40			
Premix ^1)^	5.00	5.00			
De-mold agent	0.20	0.20			
Total	100.00	100.00			

TMR, total mixed ration; EG, early gestation; LG, late gestation. ^1)^ Premix provide for per kilogram TMR: VA 30000 IU, VD 10000 IU, VE 100 mg, Fe 90 mg, Cu 12.5 mg, Mn 50 mg, Zn 80 mg, Se 0.3 mg, I 0.8 mg, Co 0.5 mg. The premix was provided by Nongbolier Technology Co., Ltd. (Beijing, China). ^2)^ Metabolizable energy value was calculated according to the NRC, unit: Mcal/kg.

**Table 2 animals-10-00433-t002:** Folate and Hcy concentrations in the serum from mating to parturition.

Item	Time	LZ		D			SEM	*P*-Values				
	(d)	TW	TR	CON	F16	F32		LZ	D	I ×	L	Q
**Folate (ng/ml)**	1	18.11	18.67	14.51 ^c^	18.09 ^b^	22.57 ^a^	0.63	0.325	<0.001	0.889	<0.001	0.378
	30	17.19	17.23	13.35 ^c^	17.24 ^b^	21.04 ^a^	0.58	0.923	<0.001	0.896	<0.001	0.895
	90	14.7	13.87	10.88 ^c^	14.86 ^b^	17.11 ^a^	0.46	0.01	<0.001	0.91	<0.001	0.019
	140	17.44	17.97	14.25 ^c^	18.52 ^b^	20.35 ^a^	0.5	0.241	<0.001	0.967	<0.001	0.006
	P	18.7	19.35	15.70 ^c^	19.90 ^b^	21.48 ^a^	0.48	0.23	<0.001	0.984	<0.001	0.017
**Hcy (μmol/L)**	30	6.98	7.04	8.48 ^a^	6.24 ^b^	6.31 ^b^	0.24	0.857	<0.001	0.746	<0.001	0.008
	90	7.48	7.96	9.12 ^a^	7.13 ^b^	6.91 ^b^	0.23	0.171	<0.001	0.855	<0.001	0.018
	140	6.88	7.03	7.91 ^a^	6.61 ^b^	6.34 ^b^	0.16	0.555	<0.001	0.986	<0.001	0.048
	P	8.04	8.22	9.89 ^a^	7.46 ^b^	7.04 ^b^	0.27	0.62	<0.001	0.972	<0.001	0.007

Hcy, homocysteine; LZ, litter size; D, diet; TW, twin born; TR, triplet born; CON, control group; F16 and F32, 16 or 32 mg·(kg.DM)^−1^ FA supplemented in the basal diets, respectively; I ×, interaction effect; L, *P* value of liner effect; Q, *P* value of quadratic effect; SEM, standard error of mean; P, postpartum. Different superscripts in the same row means significant difference.

**Table 3 animals-10-00433-t003:** The concentrations of IGF-I and GH affected by litter size and dietary FA levels (ng/mL).

Item	Time	LZ		D			SEM	*P*-Values				
	(d)	TW	TR	CON	F16	F32		LZ	D	I ×	L	Q
**IGF-I**	30	242.59	248.17	241.25	252.52	242.37	4.64	0.592	0.619	0.94	0.87	0.328
	90	263.55	273.01	252.68 ^b^	276.12 ^a^	275.99 ^a^	3	0.053	<0.001	0.611	0.001	0.023
	140	246.7	256.96	237.17	258.23	260.09	4.25	0.225	0.076	0.897	0.023	0.215
	P	265.57	272.84	255.93 ^b^	278.24 ^a^	273.43 ^a^	3.55	0.284	0.035	0.54	0.046	0.058
**GH**	30	10.41	10.33	10.39	10.2	10.53	0.22	0.876	0.846	0.924	0.571	0.691
	90	12.12	12.46	11.82	12.4	12.66	0.17	0.324	0.151	0.493	0.035	0.56
	140	12.12	12.29	11.97	12.44	12.21	0.2	0.673	0.655	0.304	0.44	0.307
	P	13.19	13.52	12.57	13.98	13.51	0.23	0.482	0.061	0.843	0.092	0.044

IGF-I, insulin-like growth factor-I; GH, growth hormone; LZ, litter size; D, diet; TW, twin born; TR, triplet born; CON, control group; F16 and F32, 16 or 32 mg·(kg.DM)^−1^ FA supplemented in the basal diet, respectively; I ×, interaction effect; L, *P* value of liner effect; Q, *P* value of quadratic effect; SEM, standard error of mean; P, postpartum. Different superscripts in the same row means significant difference.

**Table 4 animals-10-00433-t004:** The change of IGF-I and GH in serum over time of pregnancy (ng/mL).

Item	Groups	Time (d)				SEM	*P*-Values		
		30	90	140	P		T	L	Q
**IGF-I**	TW-CON	239.09	250.93	233.4	257.83	5.45	0.414	0.447	0.571
	TW-F16	247.34	271.3	250.51	271.66	4.48	0.082	0.168	0.865
	TW-F32	241.33	268.41	256.2	267.2	4.83	0.175	0.127	0.391
	TR-CON	243.4	254.42	240.94	254.02	5.87	0.83	0.758	0.938
	TR-F16	257.70 ^b^	281.04 ^a^	265.94 ^ab^	284.83 ^a^	3.76	0.034	0.043	0.737
	TR-F32	243.39 ^b^	283.57 ^a^	263.98 ^ab^	279.66 ^a^	4.83	0.003	0.012	0.111
**GH**	TW-CON	10.54	11.45	11.51	12.43	0.27	0.109	0.02	0.977
	TW-F16	10.12 ^c^	12.15 ^b^	12.28 ^b^	13.66 ^a^	0.33	<0.001	<0.001	0.447
	TW-F32	10.57 ^b^	12.76 ^a^	12.56 ^a^	13.49 ^a^	0.32	0.003	0.001	0.199
	TR-CON	10.22 ^b^	12.19 ^a^	12.42 ^a^	12.71 ^a^	0.34	0.03	0.007	0.132
	TR-F16	10.28 ^c^	12.65 ^b^	12.59 ^b^	14.31 ^a^	0.37	0.001	<0.001	0.54
	TR-F32	10.49 ^b^	12.55 ^a^	11.86 ^ab^	13.53 ^a^	0.36	0.01	0.004	0.744

IGF-I, insulin-like growth factor-I; GH, growth hormone; P, postpartum; SEM, standard error of mean; T, time effect; L, *P* value of liner effect; Q, *P* value of quadratic effect. TW-CON, TW-F16 and TW-F32, ewes lambing twins and fed 0, 16, or 32 mg·(kg.DM)^−1^ FA in the basal diet, respectively; TR-CON, TR-F16 and TR-F32, ewes lambing triplets and fed 0, 16, or 32 mg·(kg.DM)^−1^ FA in the basal diet, respectively. N = 5 for TR-CON, and n = 6 for other treatments. Different superscripts in the same row means significant difference.

**Table 5 animals-10-00433-t005:** Serum immunoglobulin concentration influenced by litter size and dietary FA levels (g/L).

Item	Time	LZ		D			SEM	*P*-Values				
	(d)	TW	TR	CON	F16	F32		LZ	D	I ×	L	Q
**IgG**	30	10.95	11	11.01	11.3	10.61	0.22	0.91	0.442	0.526	0.597	0.192
	90	11.94	11.91	12.16	11.9	11.71	0.21	0.944	0.74	0.897	0.422	0.944
	140	11.74	12.25	10.42 ^b^	12.68 ^a^	12.88 ^a^	0.28	0.237	0.002	0.219	<0.001	0.033
	P	13.46	14.48	12.20 ^b^	14.98 ^a^	14.73 ^a^	0.32	0.033	<0.001	0.303	0.001	0.006
**IgM**	30	1.61	1.57	1.57	1.57	1.64	0.03	0.524	0.524	0.925	0.394	0.532
	90	1.58	1.6	1.56	1.59	1.64	0.03	0.696	0.46	0.457	0.25	0.843
	140	1.63	1.64	1.64	1.62	1.64	0.03	0.917	0.963	0.979	0.919	0.762
	P	1.73	1.77	1.62 ^b^	1.83 ^a^	1.80 ^a^	0.03	0.459	0.023	0.517	0.025	0.058
**IgA**	30	0.45	0.44	0.43	0.45	0.45	0.02	0.849	0.689	0.845	0.543	0.594
	90	0.43	0.43	0.41	0.45	0.43	0.01	0.988	0.345	0.964	0.323	0.229
	140	0.45	0.44	0.42	0.46	0.46	0.01	0.856	0.087	0.879	0.035	0.337
	P	0.43	0.43	0.41 ^b^	0.44 ^a^	0.44 ^a^	0.01	0.9	0.049	0.95	0.015	0.225

IgG, immunoglobulin G; IgM, immunoglobulin M; IgA, immunoglobulin A; LZ, litter size; D, diet; TW, twin born; TR, triplet born; CON, control group; F16 and F32, 16 or 32 mg·(kg.DM)^−1^ FA supplemented in the basal diet, respectively. I ×, interaction effect; L, *P* value of liner effect; Q, *P* value of quadratic effect; SEM, standard error of mean; IgG, IgM and IgA, immunoglobulin G, M and A; P, postpartum. Different superscripts in the same row means significant difference.

**Table 6 animals-10-00433-t006:** Variance of immunoglobulin in serum from 30 days after mating to parturition (g/L).

Item	Groups	Time (d)				SEM	*P*-Values		
		30	90	140	P		T	L	Q
**IgG**	TW-CON	10.67	12.13	10.67	12.22	0.3	0.089	0.217	0.934
	TW-F16	11.62 ^b^	11.83 ^b^	12.4 ^b^	14.26 ^a^	0.32	0.005	0.001	0.11
	TW-F32	10.55 ^b^	11.88 ^b^	12.14 ^b^	13.92 ^a^	0.37	0.004	0.001	0.691
	TR-CON	11.35	12.2	10.18	12.17	0.35	0.156	0.876	0.388
	TR-F16	10.98 ^c^	11.98 ^bc^	12.96 ^b^	15.71 ^a^	0.45	<0.001	<0.001	0.105
	TR-F32	10.66 ^c^	11.55 ^c^	13.62 ^b^	15.55 ^a^	0.48	<0.001	<0.001	0.345
**IgM**	TW-CON	1.6	1.59	1.65	1.65	0.03	0.893	0.522	0.995
	TW-F16	1.6	1.54	1.61	1.77	0.04	0.143	0.073	0.131
	TW-F32	1.65	1.62	1.63	1.77	0.04	0.626	0.378	0.382
	TR-CON	1.55	1.52	1.64	1.6	0.03	0.691	0.455	0.937
	TR-F16	1.53 ^b^	1.64 ^b^	1.63 ^b^	1.89 ^a^	0.04	0.01	0.003	0.268
	TR-F32	1.64	1.66	1.64	1.83	0.03	0.1	0.046	0.185
**IgA**	TW-CON	0.44	0.42	0.43	0.41	0.01	0.687	0.338	0.912
	TW-F16	0.45	0.45	0.46	0.44	0.01	0.838	0.837	0.566
	TW-F32	0.45	0.43	0.46	0.44	0.01	0.688	0.826	0.986
	TR-CON	0.42	0.41	0.41	0.41	0.02	0.691	0.455	0.937
	TR-F16	0.46	0.45	0.46	0.44	0.01	0.843	0.526	0.849
	TR-F32	0.44	0.44	0.47	0.44	0.01	0.769	0.787	0.668

IgG, immunoglobulin G; IgM, immunoglobulin M; IgA, immunoglobulin A. P, postpartum; SEM, standard error of mean; T, time effect; L, liner effect; Q, quadratic effect. TW-CON, TW-F16 and TW-F32, ewes lambing twins and fed 0, 16, or 32 mg·(kg.DM)^−1^ FA in the basal diet, respectively; TR-CON, TR-F16 and TR-F32, ewes lambing triplets and fed 0, 16, or 32 mg·(kg.DM)^−1^ FA in the basal diet, respectively. IgG, IgM and IgA, immunoglobulin G, M and A. N = 5 for TR-CON, and n = 6 for other treatments. Different superscripts in the same row means significant difference.

**Table 7 animals-10-00433-t007:** Components of colostrum and normal milk.

Items	Time	LZ		D			SEM	*P*-Values				
		TW	TR	CON	F16	F32		LZ	D	I ×	L	Q
**Fat** **(%)**	C	12.06	11.92	11.47	12.15	12.24	0.27	0.645	0.570	0.978	0.287	0.609
	15 d	5.27	5.48	4.89	5.52	5.70	0.19	0.671	0.218	0.968	0.076	0.569
Protein (%)	C	13.46	13.23	12.68 ^b^	13.72 ^a^	13.64 ^a^	0.16	0.497	0.046	0.865	0.031	0.096
	15 d	5.59	5.47	5.26	5.66	5.67	0.10	0.496	0.098	0.845	0.042	0.242
Lactose (%)	C	3.15	3.27	3.16	3.21	3.26	0.12	0.678	0.963	0.836	0.578	0.869
	15 d	5.67	5.52	5.49	5.69	5.61	0.04	0.053	0.110	0.792	0.240	0.096
BUN (mg/dL)	C	4.81	4.74	4.91	4.71	4.71	0.38	0.693	0.619	0.955	0.313	0.512
	15 d	5.84	5.88	6.16	5.64	5.79	0.16	0.911	0.468	0.874	0.231	0.274

TW, twin born; TR, triplet born; CON, control group; F16 and F32, 16 or 32 mg·(kg.DM)^−1^ FA supplemented in the basal diet, respectively; LZ, litter size; D, diet; I ×, interaction effect; L, *P* value of liner effect; Q, *P* value of quadratic effect; SEM, standard error of mean; C, colostrum; BUN, blood urea nitrogen. N = 5 for TR-CON, and n = 6 for other treatments. Different superscripts in the same row means significant difference.

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
