# Peer review of "Effect of Dietary Folic Acid Supplementation during Pregnancy on Blood Characteristics and Milk Composition of Ewes"

_animals, 2020, doi:10.3390/ani10030433_

Round 1

Reviewer 1 Report

The manuscript is original and interesting for the scientific sector. 

The experimental plan seem appropriate.

Minor suggestions should be followed befor acceptance for publication

line 68 - please report ethical approval code

line 138 - please replace "of" with "on" 

table 7 - this table describes interaction vetween LZ and D, however this is never statistically. Therefore, I suggest to organize table 7 as the provious tables.

line 242 - please remove "While"

line 246 - please replace "with or without" with "regardless"

line 288 - please remove that from the sentence "to lambing that may...." 

Author Response

Comments and Suggestions for Authors

The manuscript is original and interesting for the scientific sector. 

The experimental plan seem appropriate.

Minor suggestions should be followed before acceptance for publication

line 68 - please report ethical approval code

Response1: We have provided the ethical approval code (Permit number: DK996) in line 68.

line 138 - please replace "of" with "on" 

Response 2: The word “of” was replaced by “on” according to the reviewer’s suggestion in line 138.

table 7 - this table describes interaction between LZ and D, however this is never statistically. Therefore, I suggest to organize table 7 as the previous tables.

Response 3: This table has revised as suggested by the reviewer (line 231-232).

line 242 - please remove "While"

Response 4: We deleted the word “While” according to the reviewer’s comment (line 247).

line 246 - please replace "with or without" with "regardless"

Response 5: We replaced “with or without” by “Regardless of” according to the reviewer’s suggestion (line 251).

line 288 - please remove that from the sentence "to lambing that may...." 

Response 6: We removed the word “that” from the sentence (line 292-294).

Reviewer 2 Report

The objective of this work was to verify the impact of folic acid supplementation on the biochemical parameters of immunology and the composition of milk from multiple gestation ewes. The manuscript adds further information on the use of folic acid in the bibliography already present on the subject in ruminants and not. Methodology, experimental design, results and conclusions are correctly presented.

Author Response

Comments and Suggestions for Authors

The objective of this work was to verify the impact of folic acid supplementation on the biochemical parameters of immunology and the composition of milk from multiple gestation ewes. The manuscript adds further information on the use of folic acid in the bibliography already present on the subject in ruminants and not. Methodology, experimental design, results and conclusions are correctly presented.

Response 1: Thanks very much for the reviewer’s comments and recognition, we will continue to work hard.

Reviewer 3 Report

Reviewed manuscript "Effect of dietary folic acid supplementation during pregnancy on blood characteristics and milk composition of ewes " (animals-729713) contains the results of very interesting research work of scientific and practical significance. The experiment was planned properly and carried out on sufficiently numerical material, kept on the professional farm.

Statistical analysis of the obtained results is correct.

Tables presented the results and statistical data were constructed properly.

The discussion was carried out properly and the literature used in this part of the manuscript was chosen accordingly.

However, it contains several inaccuracies:

- did the author consider other aspects? - stress factors which influence on animals? were they identical for all animals?

- please describe in more detail about permission of the ethics committee for the research

- it would be good to add the data on the premixture - the producer ect.

In summary - the manuscript contains valuable results and should be publishing in Animals.

Author Response

Comments and Suggestions for Authors

Reviewed manuscript "Effect of dietary folic acid supplementation during pregnancy on blood characteristics and milk composition of ewes " (animals-729713) contains the results of very interesting research work of scientific and practical significance. The experiment was planned properly and carried out on sufficiently numerical material, kept on the professional farm.

Statistical analysis of the obtained results is correct.

Tables presented the results and statistical data were constructed properly.

The discussion was carried out properly and the literature used in this part of the manuscript was chosen accordingly.

However, it contains several inaccuracies:

- did the author consider other aspects? - stress factors which influence on animals? were they identical for all animals?

Response 1: In order to avoid the influence of stress factors on the experiment, except for the difference of dietary folate levels, the rest operations and treatments of all the animals were consistent. In addition, the possible stress behaviors were minimized throughout this experiment.

- please describe in more detail about permission of the ethics committee for the research

Response 2: We provided the permit number in the revised manuscript according to the reviewer’s comment (line 68).

- it would be good to add the data on the premixture - the producer ect.

Response 3: We supplemented information about the producer in the revised manuscript as following: “The premix was provided by Nongbolier Technology Co., Ltd (Beijing, China)”. (line 92)

In summary - the manuscript contains valuable results and should be publishing in Animals.

Response 4: Thank you very much for your comments and suggestions.